# multiSLIDE is a web server for exploring connected elements of biological pathways in multi-omics data

Soumita Ghosh [1], Abhik Datta [2,3] & Hyungwon Choi [1✉]

Quantitative multi-omics data are difficult to interpret and visualize due to large volume of data, complexity among data features, and heterogeneity of information represented by different omics platforms. Here, we present multiSLIDE, a web-based interactive tool for the simultaneous visualization of interconnected molecular features in heatmaps of multi-omics data sets. multiSLIDE visualizes biologically connected molecular features by keyword search of pathways or genes, offering convenient functionalities to query, rearrange, filter, and cluster data on a web browser in real time. Various querying mechanisms make it adaptable to diverse omics types, and visualizations are customizable. We demonstrate the versatility of multiSLIDE through three examples, showcasing its applicability to a wide range of multi-omics data sets, by allowing users to visualize established links between molecules from different omics data, as well as incorporate custom inter-molecular relationship information into the visualization. Online and stand-alone versions of multiSLIDE are available at https://github.com/soumitag/multiSLIDE.

[1] Department of Medicine, Yong Loo Lin School of Medicine, National University of Singapore, Singapore, Singapore. [2] Centre for BioImaging Sciences, National University of Singapore, Singapore, Singapore. [3] Department of Biological Sciences, National University of Singapore, Singapore, Singapore. ✉email: hyung_won_choi@nus.edu.sg

The innovations of omics technologies such as massively parallel sequencing and mass spectrometry have made multi-omics analysis a routine practice in cell biology studies and in clinical applications[1]. However, multi-omics data sets are challenging to analyze not only because of the expanding dimensionality of data, but also because of the complexity that comes from the interconnected nature of multiple high-dimensional data sets. In traditional omics data analysis work-flow, the data analysis often depended on considerable reduction of data using statistical filters or abstraction via projection into a low-dimensional space for visualization and interpretation. Although data reduction is unavoidable for effective presentation of the high-dimensional data, the dependence on reduction and abstraction inevitably causes scientists to miss meaningful fraction of data features that fail to pass such filters. Therefore, there is a need to enable holistic exploration of unfiltered data prior to any statistical analysis. Easy-to-use, interactive visualization tools play an essential part in facilitating the unbiased exploration.

There are a handful of bioinformatics tools for multi-omics data visualization in the current literature. Open-source, data-rich web resources such as cBioPortal[2], UCSC Xena[3], and LinkedOmics[4] provide web-interfaces for query-based exploration and visualization of oncogenes from fixed data sources such as large-scale cancer cohorts of TCGA[5] and METABRIC[6]. Pathway-based visualizations, such as PaintOmics3[7], Escher[8], PathVisio[9], and network-based visualizations such as Cytoscape[10], 3Omics[11], and MONGKIE[12], are popular options for summarizing the complex interconnections and dynamics between biomolecules in a single snapshot. These methods are focused on either displaying quantitative values for a small number of select markers or visualizing overall trends at an abstract level such as pathways or networks.

Few existing tools directly visualize the quantitative data across all omics modalities in an intuitive manner and at a legible scale[13]. The ability to inspect the trends of individual molecular features at multiple molecular levels at once is critical for holistic understanding of multi-omics data sets, especially when molecular changes are discordant between different molecular levels. In large-scale multi-omics studies, we often find that the biological samples with the same phenotype can have large variations in their molecular profile between different omics modalities. These intra-sample variations should be accounted for before drawing biological inference from statistical models for multi-omics data[14–18].

To fill this gap, we have developed multiSLIDE, an interactive heatmap visualization tool for easy exploration of multi-omics data. Through multiSLIDE, we provide an interactive web browser interface to explore unfiltered and filtered multi-omics data through keyword-based queries. Quantitative molecular data are best represented using heatmaps[19,20], and multiSLIDE visualizes the queried fraction of the multi-omics data sets in separate heatmaps in one screen, with all panels synchronized with one another and with lines connecting related measurements to highlight the interconnectivity. In summary, the tool visualizes quantitative multi-omics data, narrowed down to a reasonable scale by keyword searches according to the user's hypothesis.

We demonstrate the visualization functionalities using three example studies with multi-omics data sets. The first dataset comes from a study profiling the time course mRNA and protein expression in HeLa cells undergoing unfolded protein response (UPR) in the endoplasmic reticulum (ER)[21]. The second data set presents connected visualization of phosphoproteome and proteome data, hierarchically linking phosphorylation sites (phosphosites) to the abundance data of their parent proteins, or linking kinase protein abundance with the phosphorylation sites on substrate proteins[22]. In the last example, we visualize the measurements of circulating microRNAs and proteins in human plasma samples of obese insulin resistant (IR) subjects and lean insulin sensitive (IS) subjects, connecting 3′ UTR sequence scan-based map of microRNAs and proteins of their target genes[23]. In combination, these examples demonstrate the versatility of multiSLIDE for visualizing key segments of multi-omics data through keyword searches and user-specified options.

## Results

**Functionalities of multiSLIDE web server.** multiSLIDE visualizes the quantitative data for molecules relevant to the keywords through heatmaps, with molecular relationship or interactions indicated by lines connecting between omics modalities. Keywords can be pathways, Gene Ontology (GO) terms, and gene identifiers, e.g., immune response, insulin signaling, or AKT1. Given keywords, multiSLIDE retrieves all matching pathways, gene ontologies, and genes (or other individual molecules), and the user selects search results that are of interest to them. multiSLIDE visualizes the quantitative data for the retrieved molecules as heatmaps, simultaneously for all input omics platforms. Users can apply additional statistical filtering on the retrieved data to narrow down to differentially expressed molecules between phenotypic groups using built-in parametric and non-parametric statistical tests. To control false discovery rate (FDR), multiSLIDE offers the Benjamini–Hochberg procedure[24]. In the procedure, we remark that the background list (all hypotheses) consists of the molecules selected by the user for visualization in the given instance of multiSLIDE query, not all molecules in the entire dataset. multiSLIDE also offers two distinct modes of data clustering, namely synchronized and independent modes. The synchronized mode rearranges the queried molecules in individual heatmaps based on the clustering of one omics data (anchor data), while the independent mode clusters each omics data separately.

For some omics modalities, it is possible to summarize the data at whole gene level. For instance, when visualizing transcriptome and proteome, both datasets have measurements at the gene level. In such cases, a linker can synchronize the relationship between the heatmaps, resulting in an one-to-one mapping between molecules. In this instance, a linker is simply a molecular identifier that is common between datasets. In the absence of shared molecular identifiers between two omics data, any pair of molecules in different omics data are considered independent. This independent mode is necessary, for instance, when visualizing microRNA and protein data, where molecular identifiers do not overlap.

It is often necessary to visualize quantitative data at a resolution deeper than genes. For instance, some data are summarized by genomic location or by mRNA transcripts rather than the whole gene, or by peptides rather than whole protein. These nested identifiers, unless specifically linked, are distinct across omics modalities and may have many-to-many relationships. multiSLIDE can automatically recognize nested identifiers and group them for visualization. For instance, in the second case study, the phosphorylation site-level data has nested identifiers: proteins and their amino acids amenable to a given type of posttranslational modification, e.g., serine, threonine, and tyrosine residues for phosphorylation (p-sites). For this dataset, when features are ordered by proteins, multiSLIDE performs two-level hierarchical clustering. The order of proteins is determined by clustering the average peptide intensities (default choice) of all p-sites in the protein, and the order of p-sites within each protein is determined by independently clustering the data of the p-sites. The user can choose a summary statistic (average, maximum, minimum, or sum) to compute protein-level summaries.

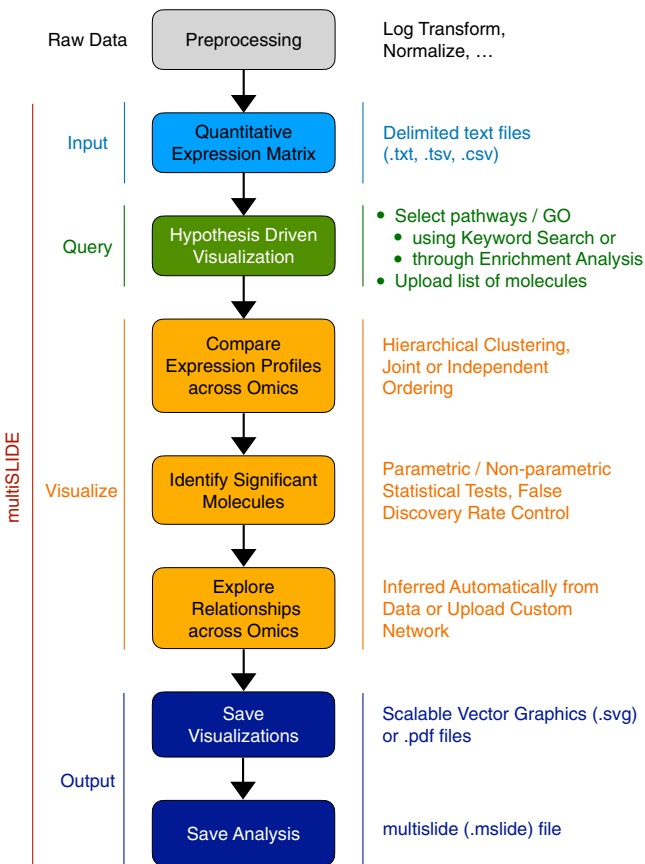

**Fig. 1 Visualization workflow of multiSLIDE.** Inputs to multiSLIDE are preprocessed quantitative expression profiles, formatted as delimited text files, with a separate file for each omics data. Users can select features to visualize, using keyword-based search, using the "Upload Pathways" option, or through enrichment analysis. Users can interact with the selected data using the many options for ordering/clustering of molecules and samples, as well as the customizable filtering of molecules based on differential expression levels. Once the exploration of the data reveals interesting patterns, users can save the visualizations as scalable vector graphics (SVG) or PDF files. The analysis workspace can also be saved as a.mslide file, retaining user selections and interactions, for sharing among collaborators. Snapshots of the visualization interface corresponding to this workflow are presented in Supplementary Fig. 1.

multiSLIDE can also infer the relationship between omics modalities when datasets have shared identifiers. For instance, when visualizing transcriptomic and proteomic data, if the transcriptomics dataset contains gene symbols and the proteomic dataset contains Uniprot accession identifiers, multiSLIDE internally maps both identifiers to Entrez, and connect genes and proteins that have the same Entrez identifiers. We refer to these standard identifiers, which can be mapped to Entrez, as "linkers".

When visualizing omics modalities where the resident molecules are mutually exclusive, e.g., proteome and metabolome, multiSLIDE allows users to input the inter-omics relationship data for linked visualization. Further, externally curated biological networks, such as transcription factor (TF) regulatory networks[25–30] and kinase-substrate networks[31–33], can be integrated and visualized through linkers in multiSLIDE.

Figure 1 illustrates a typical workflow in multiSLIDE. The structure of delimited input text files, the various querying mechanisms, and the output figure and analysis file options, are discussed in the Visualization Workflow section of Methods. The workflow is also described in the context of the multiSLIDE interface in Supplementary Fig. S1.

**Case Study I: Dynamic transcriptome and proteome in HeLa cells during ER stress**. In the first case study, we visualize the preprocessed and filtered data from Cheng et al.[21] in multiSLIDE using search keywords chosen from the original paper. The authors characterized acute transcription regulation and delayed translation control for genes involved in the following functions: UPR, translation attenuation, ER-associated protein degradation, and cellular apoptosis. A direct consequence of ER stress is aggregation of misfolded and unassembled proteins in the organelle. As a survival mechanism following the loss of homeostasis, the ER responds by increasing protein folding capacity. UPR is involved in extensive reprogramming of the transcription and translation regulation[34–36]. Activated UPR initiates adaptive stress response to regulate downstream effectors, and switches on/off transcription regulation and protein synthesis to restore ER homeostasis[37,38]. To visualize this without filtering out any genes, we searched keywords "unfolded protein response" and "endoplasmic reticulum" to retrieve all related pathways and GO terms. Genes for pathways and GO terms selected from the search results were retrieved and visualized for both transcriptome and proteome data.

In addition, prior to multiSLIDE visualization of matching mRNAs and proteins for 1237 genes, the mRNA data was separately inspected in the context of the whole transcriptome (16,704 genes) using a related single-omics data visualization tool SLIDE[39] (see Methods and Supplementary Fig. S2). This global, unfiltered view shows the three phases of ER stress response characterized by Cheng et al.: early phase (<2 h), intermediate phase (2–8 h), and late phase (>8 h). The transcriptome regulation suggests a spike-like pattern in the transition from the early phase to the intermediate phase of the response, peaking in the intermediate phase before returning to original levels or converging to new equilibrium states different from 0 h in the late phase.

In Fig. 2a, we show heatmap visualizations of the selected UPR and ER stress-related genes from the dual-omics data in multiSLIDE. We clustered the genes by hierarchical clustering of the mRNA data with Euclidean distance and average linkage, which automatically synchronizes the order of display in the protein data. The lines between the two omics data connect the mRNA and protein molecules from the same genes. In the web interface, the user can click on specific genes in one omics data to highlight their corresponding molecules in the other omics data. The user can rearrange this visualization by performing further clustering on the protein data (Fig. 2b), generating an "independent" clustering outcome. As the connecting lines follow the original map throughout these operations, the user can easily track the concordance or discordance of mRNA and protein expression patterns across the samples (time points here). These dynamically changing visualization instances clearly show two key observations: (i) the time course patterns in these selected genes are highly consistent between replicates in some genes, and (ii) the time course patterns are discrepant for many genes, and they highlight a 2 h time gap in response time in some genes such as HSPA5 (BiP/GRP-78) and protein disulfide isomerases (PDIA3, PDIA4, and PDIA6) between the mRNAs and the proteins, suggesting these HeLa cells underwent considerable cellular reorganization at 2 h post treatment and subsequent control of protein translation of key ER stress genes.

For the genes reported by the keyword searches, the independent clustering of the two omics data show that some

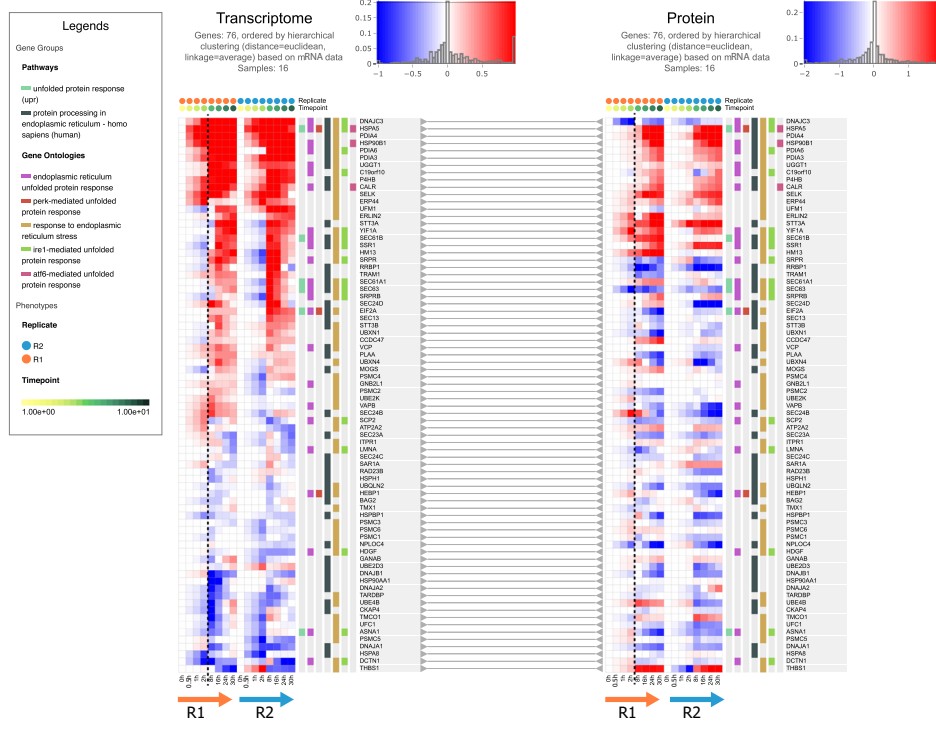

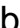

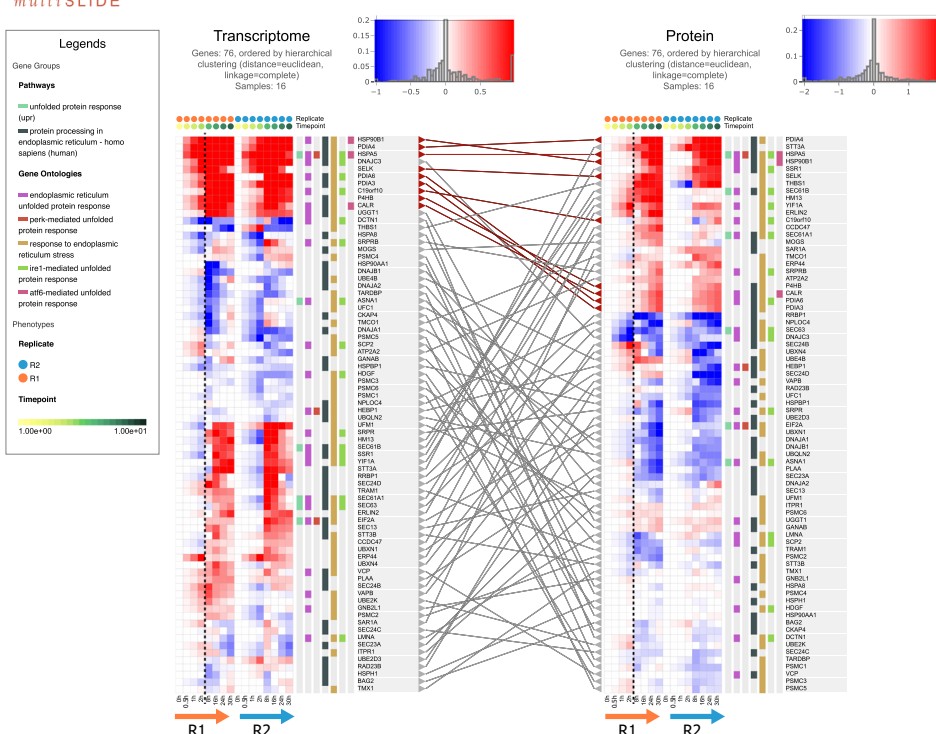

**Fig. 2 Visualization of unfolded protein response in mammalian cells responding to stress.** mRNA and protein-level expressions, across eight time-points (0, 0.5. 1, 2, 8, 16, 24, and 30 h after treatment) and two replicates, are jointly visualized in multiSLIDE to understand the dynamics of UPR under ER stress. The Legends panel, on the left, enumerates the selected GO terms and pathways. The colored tags in vertical tracks alongside the heatmap indicate associations between genes and GO terms/pathways. Panels **a** and **b** represent the two modes of visualization, synchronized and independent (unsynchronized), respectively. In the synchronized clustering mode, the same order of genes is applied to both the mRNA and protein levels. In the independent clustering mode, mRNA and protein data were clustered independently, using Euclidean distance and complete linkage.

genes are transcriptionally regulated as early as 0.5 h. Not surprisingly, the most pronounced gene expression response was observed for HSPA5, the master sensor of misfolded proteins in ER[36], as well as heat shock protein 90 beta family member 1 (HSP90B1), and PDIs. In contrast to the transcription regulation of these genes, the corresponding protein abundances do not increase until after 2 h (top portion of Fig. 2a), which corresponds to the time point until which the cells remained under cell cycle arrest.

In sum, jointly visualizing the mRNA and protein expression data in a time-course dependent manner helps the user to dissect the dynamic response stages around the prior knowledge of UPR. Clustering genes at the mRNA level and applying the same ordering at the protein level helped visualize whether the clusters propagate between the two omics levels.

**Case Study II: Proteome and phosphoproteome visualization in an ovarian cancer cohort.** We next visualized high-grade serous ovarian carcinoma data[22] from mass spectrometry-based proteomics and phosphoproteomics experiments conducted by the Clinical Proteomics Tumor Analysis Consortium (CPTAC). A total of 67 tumor samples with proteomic data (3329 unique proteins) and phosphoproteomic data (5746 p-sites) were visualized using multiSLIDE (see Methods).

In this example, we demonstrate the custom network feature of multiSLIDE. By connecting the abundance of whole proteins with individual amino acid-level phosphorylation data, we pursue two visualization objectives. First, we visualize the relationships between the abundance of proteins along with the site-specific phosphorylation levels in the same proteins. As the protein identifiers are present in both datasets, multiSLIDE automatically derives the relationship. Second, we use multiSLIDE to simultaneously visualize the kinase proteins from the proteomics data and the substrate sites from the phosphoproteome data, with lines connecting known kinase-substrate pairs. In this instance, we uploaded the "custom" network from externally curated[31–33] kinase-substrate map to the tool as a user (see Methods for details).

We remark that this data example is different from the first example in terms of the mapping of identifiers between the two omics data. In the previous example of mRNA and protein data, each gene appeared at both molecular levels, and thus there was a one-to-one mapping between the omics data sets. By contrast, the proteome and phosphoproteome data are reported at different granularities—multiple phosphorylation sites reside in a protein sequence. This creates one-to-many mapping between the proteome and phosphoproteome.

In the original analysis of the data by Zhang et al., the authors identified five proteome-based molecular subtypes: differentiated, immunoreactive, proliferative, mesenchymal, and stromal, with enrichment of distinct pathways in the discriminating protein signatures. In multiSLIDE, we initiate visualization by searching the keywords: DNA replication, cell–cell communications, and complement cascade, corresponding to the authors' enriched pathways[22]. Applying one-way ANOVA ($p$ value ≤0.05) and a multiple testing-corrected significance threshold of 5% FDR (Benjamini–Hochberg procedure, built-in feature), there were a total of 610 proteins and 490 phosphorylation sites (p-sites) that are statistically significant in the comparisons of the subtypes (Supplementary Fig. S3).

Next, we looked more closely at the GO term "extracellular matrix" (ECM) alone. Using one-way ANOVA ($p$ value ≤0.05 followed by FDR 10%), we found 155 proteins and 116 phosphosites that are differentially expressed (Supplementary Fig. S4). Unsynchronized, independent hierarchical clustering of

the proteins and the p-sites, using Euclidean distance and average linkage, shows that a number of ECM proteins are elevated in the mesenchymal and stromal subtypes. Further, a subset of ECM proteins is dominantly upregulated in the stromal subtype and multiple ECM proteins, including plectin (PLEC), lamin A/C (LMNA), filamin A (FLNA), and vimentin (VIM), and they have multiple phosphorylated sites. Vimentin is an important marker for the epithelial-mesenchymal transition in tissues (EMT), a phenomenon where cells undergo transition from epithelial to mesenchymal phenotype, ultimately leading to cancer metastasis[40]. The visualization immediately shows that p-sites in PLEC, LMNA, FLNA, VIM also show consistent subtype specificity in the mesenchymal and stromal subtypes, suggesting possibility of these protein modifications being a hallmark of those subtypes. Studies have shown that vimentin, a type III intermediate filament (IF) protein, is hyperphosphorylated during mitosis by serine/threonine protein kinases involved in cell cycle which promotes the disassembly of its filamentous structure[41,42]. Filamin A, an actin binding protein, is also phosphorylated at multiple sites by different protein kinases. To understand kinase-dependent phosphorylation pathways, we next visualize the kinases present in the proteomic data jointly with the respective substrates in the phosphoproteome data.

The human kinome consists of ~518 kinases[43], among which 83 were present in the current proteome dataset. In our curated kinase-substrate map, we found these 83 kinases phosphorylates 8269 substrates. Among these 8269 substrates, 454 were present in the current phosphoproteome data. In multiSLIDE, users can select molecules to visualize by using the search functionality as described before, or through enrichment analysis (see Methods), or by uploading customized subsets of molecules. Here, using the third option, we jointly visualized 83 protein kinases and 454 substrates (Supplementary Fig. S5). At the protein level, the calcium/calmodulin-dependent protein kinases (CAMK2B, CAMK2G, CAMK2D, and CAMK2A) are upregulated in mesenchymal and stromal subtypes, whereas the CMGC kinases, consisting of cyclin-dependent kinases (CDK1, CDK2, and CDK11A) and glycogen synthase kinases (GSK3A and GSK3B) are upregulated in the proliferative subtype. These results affirm the widely known role of CDK1 and CDK2 in orchestrating mitotic progression, a phase in the cell cycle process during which protein phosphorylation is also known to peak[44].

To further investigate subtype-specific protein kinase activity, we calculated the Pearson correlation coefficients between the subtype-specific levels of kinase abundance and substrate site-level phosphorylation, outside of multiSLIDE. Supplementary Fig. S6 shows the histograms of calculated correlation coefficients for each subtype. The subtypes: immunoreactive, proliferative and stromal have relatively greater numbers of highly positively correlated (≥0.8) kinase-phosphosite pairs. In the "proliferative" subtype, among these kinase-substrate p-site pairs, those pairs upregulated in both omics were visualized in multiSLIDE (Fig. 3), mimicking the kinase-substrate enrichment analysis[45]. In addition, the Uniform Manifold Approximation and Projection (UMAP) visualization of the entire proteome and phosphoproteome data, performed outside multiSLIDE and shown in Fig. 3b, c respectively, also revealed stratification of the proliferative subtype patients, driven by the portion of the data visualized above[46].

Here, we uploaded the kinase-substrate relationships into multiSLIDE using the network upload feature, which allows users to visualize externally created inter-omics connections. The lines from CDK1 and CDK2 point to all their known substrates. We see that most of the substrates of CDK1 and CDK2 show elevated phosphorylation levels in the "proliferative" and the "immunoreactive" subtypes. These proteins are involved in the G1/S phase

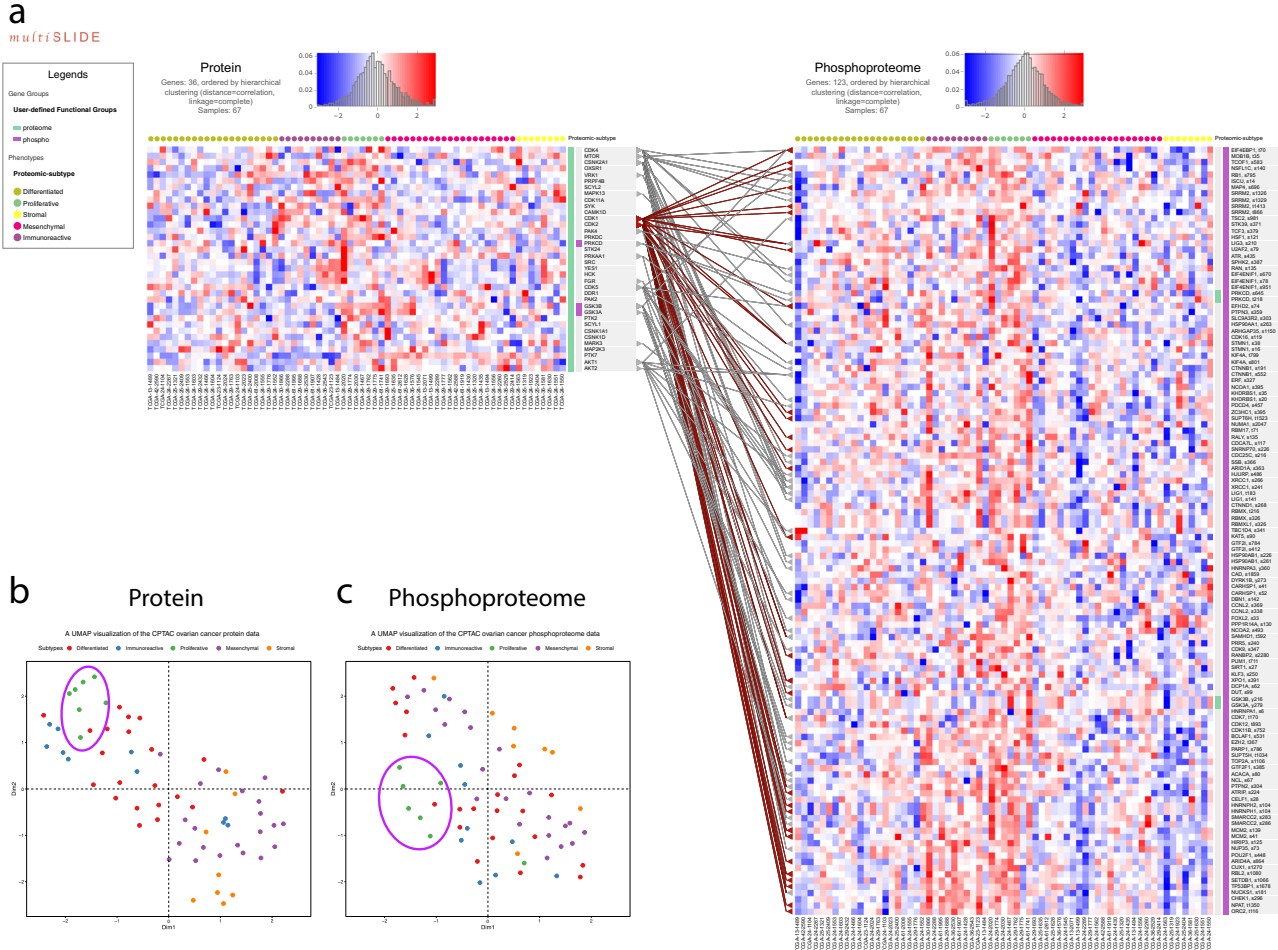

**Fig. 3 Visualization of kinase-substrate relationships in the CPTAC Ovarian Cancer data. a** Visualization of the subset of kinases-substrate pairs, which are upregulated in the proliferative subtype. The custom "Upload" option is used to select the molecules here. Kinase-substrate interactions were curated from PhosphoSitePlus[31], PhosphoNetworks[32], and a predictive network inference approach[33] to build a kinase-substrate map, which was uploaded into multiSLIDE using the upload network feature. The connecting lines show these curated relationships, with the highlighted (brown) lines connecting cyclin-dependent kinases CDK1 and CDK2 with known substrates. Supplementary Fig. S5 visualizes all the kinases-substrate pairs. **b** A UMAP visualization of the whole proteomics data for 3329 proteins. The ellipse highlights a cluster of proliferative subtype patients in the protein data. **c** A UMAP visualization of the whole phosphoproteome data for 5746 phosphosites. The ellipsis highlights a cluster of proliferative subtype patients in the phosphoproteome data.

transition which is known to be initiated by cyclin-dependent kinases[47]. The examples showcase the different querying mechanisms available in multiSLIDE, giving users the flexibility to visualize any subset of the data while retaining the relational information between omics data.

**Case Study III: Human plasma proteome and microRNAome associated with insulin resistance.** In the last example, we visualize non-matching molecular entities between omics modalities, i.e., plasma proteins and miRNAs between eight IR and nine IS subjects[23]. miRNAs are small non-coding RNAs that control the fate of target mRNAs through mRNA degradation or translation repression[48]. Specifically, by binding to the sequence motifs in the 3′ UTR region of the target gene, miRNA diverts the mRNAs away from ribosomes and thereby inhibits protein translation, or primes the mRNAs for degradation via deadenylation and decapping[49]. The integrative analysis in the original paper incorporates this negative relationship into the search to identify IR-associated plasma proteins and circulating miRNAs that are negatively correlated with the proteins. The authors made the assumptions that an elevated level of a circulating miRNA reflects increasing transcription of the miRNA in the originating

donor tissues (or organ systems) and it would have resulted in reduced secretion of the target protein into the blood as well.

What sets this visualization apart from the previous two examples is that there is no direct link between the mapping identifiers of the two omics modalities (Fig. 4). The only genome-scale relationship data between proteins and miRNAs are miRNA target sites predicted in silico, which can be achieved by a variety of computational tools[50–53]. Using multiSLIDE, we searched keywords: metabolism, metabolic pathways, inflammatory response, glucose transport, and lipid homeostasis. The original query resulted in a long list of 414 proteins and 191 miRNAs (Supplementary Fig. S7), and thus we performed hypothesis testing with FDR control ($p$ value ≤0.05 followed by FDR 5%) within multiSLIDE (Mann–Whitney $U$-test and Benjamini–Hochberg method). This functionality gives users the flexibility to adjust the size of data for display. This internal filtering produced 19 proteins and 76 miRNAs significantly different between IR and IS subjects (see Methods).

In Fig. 4, although there is variation within the IR and IS groups, hierarchical clustering using correlation distance (1 minus Pearson correlation) and complete linkage function in multi-SLIDE reveals two clear clusters in the miRNA data, separated by

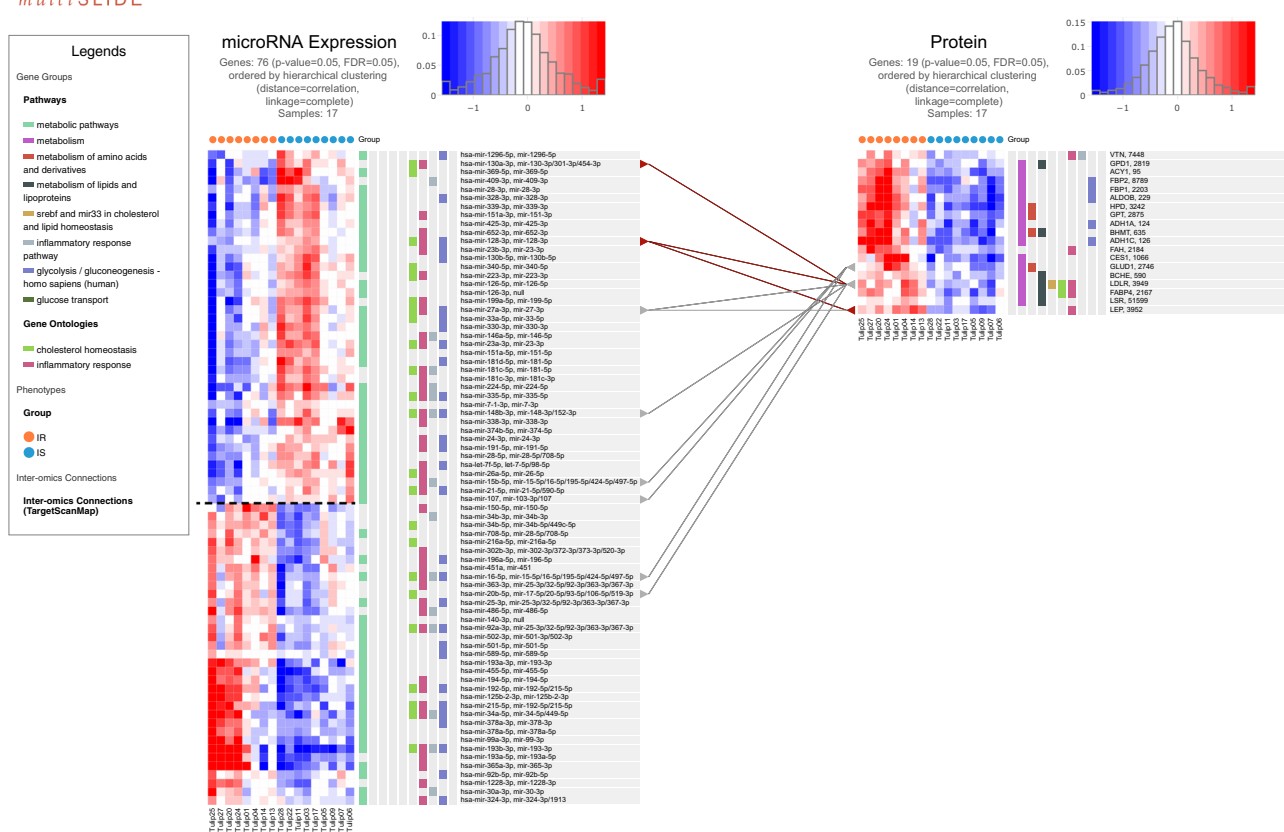

**Fig. 4 Visualization of plasma proteins and microRNAs associated with insulin resistance.** Shown in the heatmap are the molecules matched by the keywords: metabolism, inflammatory response, glucose transport, and lipid homeostasis are visualized here after filtering by Mann–Whitney U-test (p value ≤ 0.05 followed by FDR 5%). Proteins and miRNA are independently clustered using correlation distance (1 minus Pearson correlation) and complete linkage function. The relationships between miRNA family names and their target proteins are extracted from TargetScanMap[50]. The original list of 414 proteins and 69 miRNAs, before filtering, is shown in Supplementary Fig. S7.

the dotted black line in Fig. 4. At the protein level, however, we find that most plasma proteins are higher in the IR subjects than in the IS subjects, forming a homogeneous cluster. This observation prompts a reasonable speculation that the top cluster may have to do with increased secretion of the selected 19 proteins at their respective donor tissues, likely from the organ systems such as liver, adipose tissues, or kidney.

To further probe whether the proteins are potential targets of translation control by these miRNAs, in whichever tissue they originated from, we connected miRNAs and proteins in multi-SLIDE with TargetScan map[50]. Any known miRNA target protein is indicated by a connecting line in Fig. 4. Interestingly, some of the lines showed that higher circulating levels of miR-130a-3p and miR-128-3p in the IS subjects were associated with lower level of their predicted target gene, low density lipoprotein receptor (LDLR), a type-I transmembrane glycoprotein. It is widely known that LDLR plays a critical role in maintaining cholesterol homeostasis in the blood, and while insulin resistance is defined by abnormal glucose metabolism, its pathogenesis is increasingly being studied in the context of disordered lipid metabolism[54]. The negative correlation between the two miRNAs (miR-130a-3p and miR-128-3p) and the expression of LDLR in IR subjects, evident from Fig. 4, is therefore interesting. This has also been previously shown, through GWAS meta-analysis[55], where miR-128-1 was identified as a key to controlling LDL-C uptake by regulating the expression of LDLR. Also, a previous investigation of the role of miR-130a-3p suggested that its overexpression improves insulin sensitivity both in vitro and in vivo[56]. All put together, this example shows that multiSLIDE

can integrate externally curated networks in multiSLIDE through visualization of relationships between mutually exclusive omics data sets.

## Discussion

We have developed multiSLIDE, a new web-based tool for interactive heatmap-based visualization of multi-omics data. With steady growth in multi-omics experiments, it is becoming increasingly appealing to develop open-source analysis and visualization tools. Using the versatile interface of multiSLIDE, users can retrieve segments of data through keyword searches and generate interpretable visualizations, with the flexibility to control the size and readability of display contents. Existing tools for multi-omics visualization tend to focus on displaying a small number of genes within publicly available datasets, such as TCGA, or visualize patterns at abstract levels without showing the actual quantitative data. multiSLIDE addresses this gap–visualizing the relational data along with actual quantitative data[57]. We demonstrated how multiSLIDE enables targeted exploration of large multi-omics datasets within proper biological contexts. Because the architecture of multiSLIDE was built as a web-based tool, the saved contents can be opened in any other computer with a standard web browser. As such, multiSLIDE was designed to treat both the data analysis and visualization as resources to be shared and disseminated for collaborative research, a feature that is often missing in currently available tools.

multiSLIDE has limitations, nonetheless. multiSLIDE is primarily a search-driven visualization tool, which requires the users

to have prior biological hypotheses. Therefore, it may not be particularly well suited for the unrestricted exploration of whole multi-omics datasets. Such global exploration can be done using other tools such as multiSLIDE's sister tool SLIDE[39]. As an alternative querying mechanism, multiSLIDE provides the option to detect differentially expressed genes and perform enrichment analysis to identify key pathways and GO terms enriched in the user's data. From the result of enrichment analysis, users can select pathways and GO terms for visualization instead of having to provide search keywords. A full-scale global visualization of data in SLIDE can be a useful precursor to analyzing the data in multiSLIDE. In addition, multiSLIDE does not provide functionalities to preprocess and normalize data within the tool and expects the user to prepare display-ready data sets . This was an unavoidable choice as these preprocessing operations are often better handled by validated domain-specific tools.

Visualization is an important tool to inspect data, gain new insights, and communicate the findings to others, especially for the analysis of complex multi-omics data. multiSLIDE has a unique focus on visualizing raw quantitative data relevant to the biological functions and genes queried by the user and integrating information from multiple omics sources through relational information, in contrast to the abstraction or projection-based approaches commonly built in the existing multi-omics visualization strategies that often do not directly visualize the feature-level data. As such, multiSLIDE can be used as a complementary visualization tool to revisit the raw data along with other data analysis methods or global visualization tools already available in the literature.

## Methods

**Visualization workflow**. Figure 1 illustrates a typical workflow in multiSLIDE. The web-based visualization interface is shown in Supplementary Fig. S1a. Data analysis begins with the user specifying or selecting pathways, GO terms, or individual genes to visualize using one of three options: keyword search, enrichment analysis, or pathway upload (see Input Data section for details). For the first option, multiSLIDE provides an intuitive keyword-based search syntax for quickly searching multiple pathways, GO terms, and genes. The relevant genes and gene groups are selected from the search results and they are visualized, as shown in Supplementary Fig. S1b, by clicking the group names.

Once the heatmap visualization is ready, network neighbors of a target gene on protein–protein interaction (PPI) and TF regulatory networks can be added via network neighborhood search, all enabled by a simple left-click on the gene of interest. A selection of neighbors can be added to the visualization in real time. In addition, multiSLIDE also overlays the information regarding pathways and GO terms by vertical tracks next to each heatmap, highlighting the intersections (common genes) between functional groups (side bars on the right side of heatmaps, Supplementary Fig. S1a).

The scales and dynamic range of detection and quantification are different across omics platforms. As a result, direct comparison of absolute expression values is not meaningful. Customizing the graphical parameters in each omics data are therefore essential for successful visualization. Individual heatmaps are independently customizable (see heatmap settings panel, Supplementary Fig. S1c). Settings common to all heatmaps, such as zoom (or resolution) and orientation of heatmaps, are applied to all heatmaps simultaneously using the global settings panels (Supplementary Fig. S1f). multiSLIDE has no restrictions on the amount of data that can be loaded in a single snapshot. As different systems and browsers have different computing capabilities, this choice is left to the user. Using the layout options (Supplementary Fig. S1f), the size of a single snapshot can be optimized, depending on the data transfer rate between the multiSLIDE server and the browser, and the browser's latency in rendering the data.

multiSLIDE has a variety of sorting, clustering, and filtering methods to help users discover patterns in the data. Interesting genes can be hard to discern when they are incoherently mixed with other genes, particularly when visualizing large pathways or networks. With appropriate ordering of genes and samples, previously unforeseen structures in the data can emerge. Molecules can be sorted by gene groups, based on the statistical significance level in a differential expression analysis, or based on hierarchical clustering. Samples can be ordered by a combination of phenotypes or based on hierarchical clustering for interrogating the strength of phenotype–genotype associations. Hierarchical clustering can also be customized by selecting different linkage functions, distance metrics, and leaf ordering schemes.

multiSLIDE can remove statistically non-significant genes from the visualization through internal differential expression analysis in real time basis. In

large pathways and networks, a substantial number of genes may not show differential expression, and thus removing these stably expressed or non-expressed genes improves the visualization clarity. multiSLIDE automatically classifies phenotypes into one of three categories based on the data: binary, categorical, or continuous. For binary and categorical data, users can choose to perform either parametric or non-parametric tests. The parametric tests used for binary and categorical phenotypes are two-sample t-test and analysis of variance (ANOVA), respectively. The corresponding non-parametric tests are Mann–Whitney U- test and Kruskal–Wallis test, respectively. For continuous data, linear least squares regression is used. Additionally, users can perform multiple testing correction using the Benjamini–Hochberg procedure to control the FDR.

As mentioned earlier, the user can choose to apply hierarchical clustering and filtering of features either in a synchronized mode or in an independent mode. In the synchronized mode, hierarchical clustering and filtering are performed on the features of one of the datasets selected by the user, and the ordering of that dataset is used to order the features of other datasets. This mode is only meaningful when the datasets share a linker column. By contrast, in the independent mode, hierarchical clustering and filtering are applied independently for each omics data. A combination of these modes can also be used, where a subset of omics is synchronized and a subset is kept independent, by customizing the omics relationships, as shown in Supplementary Fig. S1f.

**Software architecture**. multiSLIDE is built on a distributed architecture, shown in the schematic in Supplementary Fig. S8. The server side of multiSLIDE consists of a state server, an analytics server, and a knowledge server. The client can be any modern web browser. These four components are separate applications that communicate with each other through well-defined application programming interfaces (APIs). Due to this modular design, multiSLIDE can scale to distributed multi-node environments, with many possible deployment configurations.

The state server is an HTTP server, implemented in Java, that maintains client state information, user uploaded data, and user selections. The analytics server, also an HTTP server, is implemented in Python and is the main computation engine. The knowledge server, implemented using MongoDB, manages the physical storage of curated gene annotation, regulatory networks, biological pathways and GO terms. The analytics and knowledge servers are stateless. The client interacts only with the state server by sending HTTP requests and receiving data in response in optimized JavaScript Object Notation (JSON). The client is implemented using Angular, with the data and presentation layers completely decoupled. Layouts can therefore be altered without the need to re-fetch data from the server. The visualizations are rendered using resolution independent scalable vector graphics (SVG).

**Databases: Pathways, GO terms, and molecular networks**. multiSLIDE includes comprehensive genome-scale annotations and GO databases for mouse and human, extracted using R Bioconductor[58]. The data in these R packages are well-structured and routinely used by bioinformaticians in their analyses. multiSLIDE can recognize five standard gene identifiers (linkers): Entrez, HUGO Gene Nomenclature Committee (HGNC) Gene Symbols, ENSEMBL identifiers, NCBI Reference Sequence (RefSeq) identifiers and UniProt identifiers, as well as miRNA identifiers from miRbase[59]. To facilitate pathway and GO keyword-based search, multiSLIDE includes comprehensive biological pathways obtained from ConsensusPathDB (CPDB) (http://cpdb.molgen.mpg.de/)[60,61]. Validated miRNA target interactions on pathways and GO from miRWalk2.0 (http://zmf.umm.uni-heidelberg.de/apps/zmf/mirwalk2/)[62] are also included in multiSLIDE.

Various networks indicating relationships between molecules within the same molecular level such as PPI network (within proteins), as well as networks indicating relationships between molecules at different levels such as TF regulatory networks, are also integrated in multiSLIDE, to facilitate network search. These networks come from diverse sources and are either experimentally validated relationships or putative interactions that rely on computational prediction. For instance, TargetScan (http://www.targetscan.org)[50], a database widely used to represent miRNA-mediated gene regulation, houses predicted targets of miRNAs and is included in multiSLIDE. multiSLIDE also integrates Human TF—targets network information from additional databases: TRED[25], ITFP[26], ENCODE[27], Neph2012[28], TRRUST[29], and Marbach2016[30]. Mouse TF—target network information was obtained directly from TRRUST. Physical interactions between proteins were sourced from iRefIndex (http://irefindex.org/wiki/index.php?title=iRefIndex)[63], which indexes PPI networks from a number of databases.

**Input data**. multiSLIDE provides a simple and intuitive interface for users to upload their own data and create an analysis. Each omics data should be uploaded into multiSLIDE in the form of a separate delimited ASCII text file, containing quantitative measurements across samples. These files can be created and edited using any text editor. Rows in the data file correspond to molecular features, and columns can either be vectors of identifiers or measurements. Data columns correspond to samples or experimental conditions and contain measurements in the form of counts, numerically encoded categorical data, or continuous data. Meta-data columns contain feature identifiers and can be numeric or non-numeric. In case feature identifiers are standard identifiers such as Entrez or gene symbol, they

can be tagged as such during analysis creation. multiSLIDE assumes that the raw data have already undergone preprocessing and transformation. Each column in the data file must have a unique column header.

In addition to data files, a separate sample information file containing sample attributes (e.g., clinical data or phenotypes) is also required. The sample information file should also be formatted as a delimited ASCII text file, with rows corresponding to samples and all columns, except the first column, corresponding to sample attributes or phenotypes. The first column in this file must contain sample names that are identical to the measurement column headers in data files. There are no restrictions on the number of sample attributes or phenotypes the sample information file can contain. The visualization interface allows users to select a subset of (at most five) phenotypes for visualization. The sample information file can also be used to include additional sample information such as descriptive sample names, replicate names, or time points.

The associations between any two omics can be customized by uploading a network file. These files must have at least two columns, one for each omics, with rows containing identifiers that are connected. Optionally, a third and fourth column, containing color information and a descriptor (such as a name or source), respectively, can be provided for each connection in the network file. These optional columns can be used to annotate connections, for instance, to visually categorize experimentally validated, putative, and predicted molecular interactions or interactions acquired from different external databases. Even when linker columns are available and multiSLIDE automatically infers the associations, network files can be used to override them.

**Querying mechanisms**. multiSLIDE provides three querying mechanisms for users to select a subset of molecules to visualize from the uploaded data files. The first option, keyword-based search, was discussed in the Visualization Workflow section above. The second querying mechanism is through enrichment analysis, which requires data files containing standard gene identifiers. Unlike keyword-based search, this option is useful when the user does not have a priori knowledge or benchmark pathways, GO terms, or genes of interest. multiSLIDE uses hypergeometric test to evaluate statistical significance of function enrichment in the differentially expressed molecules. To perform enrichment analysis, the user specifies a set of parameters, such as the phenotypic groups to compare and identify differentially expressed genes, the significance level, to name a few. The differential expression analysis performed prior to enrichment analysis requires the same user inputs and follows the same methodology as the significance based filtering feature described in the Visualization Workflow section. The background list used for the hypergeometric test includes all the molecules present in the selected omics dataset. The results of enrichment analysis are listed in the interface in ascending order of *p* values, where the user can click on enriched pathways or GO terms and add them to the visualization. Several options such as size of pathways, the number of differentially expressed genes in the pathway, are also available to filter enrichment analysis results. Users can also download a detailed report of the enrichment analysis results in a tabular format. Finally, the third querying mechanism is to upload a pathway file containing user-specified functionally relevant molecules. This option is for data files that contain non-standard identifiers. For example, when visualizing metabolomics data, since the map between genes and metabolites has not been fully charted by experimental means, standard shared identifiers cannot be provided. A pathway file contains four columns: functional group name, data filename, identifier name, and identifier value. Each row of the pathway file specifies molecules with identifiers matching the criteria identifier name = identifier value from the specified data filename and adds them to the functional group specified. A single pathway file can therefore be used to describe multiple groups of functionally relevant molecules.

**Output data**. multiSLIDE allows both the data analysis and the generated visualization to be shared and disseminated online for collaborative research, a feature that is often missing in existing visualization tools. In multiSLIDE, visualization in the current view can be rendered in a ready to print form, using the save visualization option. This re-renders the current view in a more compact form in a separate window in resolution independent SVG format. The print or save as options of the browser can be used to save the contents of the window as a high-resolution PDF file (Supplementary Fig. S1d).

The analysis workspace can be saved as.mslide files and shared among collaborating parties (Supplementary Fig. S1d). These files can be loaded back into a different instance of multiSLIDE running on any web browser for continued analysis. The saved analysis retains all user customizations and data selections, as well as the raw and processed data in JSON format.

**Data set I: Dynamic transcriptome and proteome in HeLa cells during ER stress**. The first case study explores the whole transcriptome data from Cheng et al[21]. (16704 genes). In the dual-omics time-course experiment, HeLa cells were treated with a sublethal dose of dithiothreitol (DTT) inducing ER stress, and then sampled at eight time points (0, 0.5, 1, 2, 8, 16, 24, and 30 h after treatment) for transcriptome and proteome profiling. In their study, a series of quality filters resulted in the final set of 1237 genes with matched mRNA and protein data.

Prior to visualization in multiSLIDE, we inspected the entire transcriptome data using SLIDE[39], a related tool that we have previously developed for full-scale single-omics data visualization. The log-transformed mRNA and protein measurements of each gene were normalized by subtracting the pretreatment measurement (0 h). This normalization was performed independently for each replicate, turning the abundance values into (log2) ratios to the baseline. The normalized data is visualized after applying hierarchical clustering in Supplementary Fig. S2. In Supplementary Fig. S2a, search tags highlight genes belonging to the GO term "endoplasmic reticulum unfolded protein response" (green bars to the right of the heatmap).

**Data set II: Proteome and phosphoproteome in CPTAC ovarian cancer cohort**. In the second case study, we visualize high-grade serous ovarian carcinoma data from mass spectrometry-based untargeted proteomics and phosphoproteomics experiments conducted by the Clinical Proteomics Tumor Analysis Consortium (CPTAC). Our visualization follows that of Zhang et al[22]. In their analysis, the authors retained the 3586 proteins out of the 9600 that were quantified in 169 tumors. For the phosphoproteome data, the authors quantified the relative abundance for 69 tumor samples. Among these, 67 samples were quantified at both omics level and were used in the final visualization. Phosphosites with more than 50% of missing data were filtered out and the remaining missing values were imputed using KNNImpute[64]. Since ischemia of the TCGA tumor samples was found to be a confounding variable that altered phosphopeptide abundance, phosphosites that were shown to be regulated in ovarian carcinoma[65] were also removed. The abundances were converted to z-scores before visualizing in multiSLIDE. A total of 16718 kinase-substrate interactions were curated from PhosphoSitePlus[31], PhosphoNetworks[32] and a predictive network inference approach[33] to build the kinase-substrate map.

**Data set III: Human plasma proteome and microRNAome associated with insulin resistance**. In the third case study, we visualize plasma proteins and miRNAs between eight IR and nine IS subjects[23]. The abundance values of 368 miRNAs and 1499 proteins were log transformed (base 2) and mean centered prior to visualization in multiSLIDE. The miRNAs were mapped to the corresponding miR Family obtained from TargetScan (http://www.targetscan.org)[50]. Predicted miR family-target genes network information were extracted from TargetScan and uploaded as a file into multiSLIDE to explore miRNA-mediated gene regulation.

**Reporting summary**. Further information on research design is available in the Nature Research Reporting Summary linked to this article.

## Data availability

The quantitative expression data used as inputs to multiSLIDE are available at: https://github.com/soumitag/multiSLIDE. The data for Case Study I is available online at https://www.embopress.org/action/downloadSupplement?doi=10.15252%2Fmsb.20156423&file=msb156423-sup-0002-DatasetEV1.xlsx. The data for Case Study II is available online at https://www.cell.com/cms/10.1016/j.cell.2016.05.069/attachment/15e46617-bec0-42cc-82fb-71f842e8aaac/mmc3.xlsx. The data for Case Study III can be found in Table S1 of https://www.frontiersin.org/articles/10.3389/fphys.2019.00379/full#supplementary-material.

## Code availability

Online version of multiSLIDE is available with multiple demo analyses that users can open with a single click and explore. Users can also upload their own data here. For continued use, users or facility managers are encouraged to install multiSLIDE on their own computers or servers using the pre-built docker image. The link to the online version and the docker image can be found at https://github.com/soumitag/multiSLIDE.[66]

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

## Acknowledgements

This work was supported in part by grants from Singapore Ministry of Education (MOE2016-T2-1-001 and MOE2018-T2-2-058 to H.C.), and National Medical Research Council of Singapore (NMRC-CG-M009 to H.C.).

## Author contributions

S.G. and H.C. conceived the idea. S.G. and A.D. designed the software architecture and implemented the tool. S.G. and H.C. performed all data analysis and S.G. prepared the final visualization. All authors wrote the manuscript. H.C. supervised the project.

## Competing interests

The authors declare no competing interests.
