## [Peer Review File · Nature Communications]

Reviewers' Comments:

Reviewer #1:

Remarks to the Author:

The tool developed by Ghosh et al. "multiSLIDE: a web server for exploring connected elements of biological pathways in multi-omics data". I'm not aware of other tools that are doing this, though there are plenty of tools to find relationships between data, and plenty of tools to do data exploration. The added value in here seems to be the integration of the heatmaps with the edge addition between omics.

Major Comments:

1. How to save the newly generated information such as multi-omics associations or pathway associations in a computationally tractable way (e.g. exporting identified interactions between omics in .CSV or .TSV formats)?
2. It would be important to show the source of a newly given annotation or interaction that is added to a dataset.

For the web interface, Demo 3 (miRNA vs Protein), with everything as default:

3. What is the criterion for showing $FDR > 0.01$ ("Enter a number between 0.01-1.0")? Is it that no test is deemed statistically significant at $FDR < 0.01$, or is this imposed by the webtool?
4. Changing FDR to 0.01, 0.05, or 0.1 leads to a bug: "Error in GlobalMapConfigServices undefined". It would be important to indicate if this is because no statistically significant test is present at these FDR, or if this is a bug in the webpage.
5. The "Show neighbours > miRNA targets" does not result in any neighbours for the following miRNAs even though an edge is drawn with the protein data ("miRNA Targets of MIRNA Found 0 neighbors, of which 0 are in the datasets."): hsa-let-7g-5p, let-7-5p/98-5p; hsa-let-7a-5p, let-7-5p/98-5p; hsa-mir-301b-3p, mir-130-3p/301-3p/454-3p.

Reviewer #2:

Remarks to the Author:

Comments to authors

In their manuscript the authors present an interactive web-based tool called multiSLIDE for visualization of multi-omics data and the interconnections between different omics modalities. Visualization is carried out with heatmaps where e.g. normalized expression values for genes and proteins can be shown side by side, where interconnections are highlighted with links between the heatmaps. The tool comes with many options to customize the visualizations, such as inclusion of genes/features based functional annotations, differential testing, enrichment analysis and custom lists. multiSLIDE has a knowledge base of known interactions across omics-modalities and it is possible for users to upload their own. The web interface also allows users to change the way heatmaps are displayed, e.g. how samples and features are ordered.

The manuscript is well written and comprehensive. Three case studies on published data have been carried out to illustrate how multiSLIDE can be used to explore patterns and relations in multi-omics data. The analysis methodology is sound and the results clearly demonstrate the usefulness of the tool. The work is novel, though multiSLIDE it is an extension of the authors' previously published tool SLIDE, which is applied to single-omics data (Ghosh et al. 2019).

Ghosh, S., Datta, A., Tan, K. & Choi, H. SLIDE--a web-based tool for interactive visualization of large-scale--omics data. *Bioinformatics* 35, 346–348 (2019).

A strength of multiSLIDE is that it can be used for very different types of (processed) omics data. In

one of the case studies the authors provide an interesting and intuitive way to explore proteome/phosphoproteome data, e.g. by visualizing proteins and their phosphosites, or kinases and their substrates. This makes multiSLIDE powerful exploration tool for multi-omics studies.

I have some concerns about the manuscript and the web tool that should be addressed before the manuscript is considered further. These are outlined below.

Major revisions

The enrichment analysis option for adding genes appears non-functional. This was tested on the demo data available in <http://137.132.97.109:56695/multislide/#/>. The following error was generated: Enrichment Analysis parameters update FAILED. Could not open analysis, perhaps session has expired. Error in do_enrichment_analysis().

To test the upload of custom data, the miRNA and protein data for case study II and III were separately uploaded to multiSLIDE. To visualize heatmaps, three and four terms were added with the Search function, respectively. Case study II: DNA replication, cell-cell communication and complement cascade. Case study III: phospholipid metabolism, inflammatory response pathway, glucose transport and lipid homeostasis. However, when clicking Apply changes, the following error message was issued for both studies: Changes could not be applied. Error in AnalysisReInitializer.undefined.

Additional details about how the enrichment analysis is performed should be provided. Which hypothesis test is used and does multiSLIDE rely on ranking genes by enrichment score (as in Gene Set Enrichment Analysis) or not? What is used as background for the test? Provided that multiSLIDE accepts pre-processed data, it is more appropriate to use all features available in the data as background, rather than the whole genome.

Minor revisions

First sentence in introduction line 31: "... technologies have made the joint use of omics..."

Results line 94: The authors mention that, if a search generates too many hits, multiple testing correction could be applied. This correction is imperative to control false discovery rate, and the decided to perform it should not be based on number of hits obtained/wanted. Ideally, multiSLIDE should always perform correction by default.

Supplementary figures S2: The authors' interpretation of the heatmaps suggest that the transcriptomics profile of late time points (> 8 hours) returns to levels at time 0. While this may be true for some genes, the majority of them appear blue/red after 30 h, implying that the expression is different from 0 h.

Results line 175-177: The authors mention here that a 2-hour time gap is observed between the changes in transcript and protein profiles. In Figure 2a, this pattern can be observed for some genes (notably the top ones in the heatmap). This is not as clear for those in the middle, and for several ones the transcript and protein levels appear anti-correlated.

Method line 367: "right-click" should be "left-click" to show network neighbors

Method line 375: "left side of heatmaps" should read "right side of heatmaps"

The paper could be strengthened further by including a brief conclusion.

Web interface

Cogwheel in Synchronize Heatmap Features appears unresponsive.

View – Show Cluster Labels and View – Reset to Default Settings seem unresponsive.

It is possible to add genes etc. to feature lists, but it is unclear how these are used.

Supplementary data

README.md is confusing, it does not seem to be in agreement with the demo datasets supplied. The README does not conform to the Code and Software Submission Checklist provided with the manuscript.

Figure legends

Literature citations are not written in superscript.

In the Fig 3 legend, PhosphoSitePlus and PhosphoNetworks are cited with references 30 and 31, but should be 31 and 32, respectively.

In the Fig 4 legend, TargetScan is cited with reference 38, which appears to be a typo. The reference used in the method section is 50.

Benjamin Ulfenborg

Point-by-point response

Reviewer #1

The tool developed by Ghosh et al. "multiSLIDE: a web server for exploring connected elements of biological pathways in multi-omics data". I'm not aware of other tools that are doing this, though there are plenty of tools to find relationships between data, and plenty of tools to do data exploration. The added value in here seems to be the integration of the heatmaps with the edge addition between omics.

Major Comments:

1. How to save the newly generated information such as multi-omics associations or pathway associations in a computationally tractable way (e.g. exporting identified interactions between omics in .CSV or .TSV formats)?

Response: To address this comment, we built in a new option "Save View" under the Save option. This feature allows users to download the list of currently visualized features and connections between them as a JSON file. Data are stored in JSON files as attribute-value pairs. The Save View option in multiSLIDE results in a JSON file with an attribute-value pair for each dataset, where the attribute is the omics type and the value is an array of the features in the current view. The JSON file also contains an attribute-value pair for each inter-omics connection in the current view. Here the attribute names are semi-colon separated pairs of omics types and the values are arrays of semi-colon separated pairs of connected features.

2. It would be important to show the source of a newly given annotation or interaction that is added to a dataset.

Response: To distinguish diverse types of interactions, we have extended the format of the inter-omics connection file. Each row in the file still specifies one interaction, and the file can have up to four columns now. The first two columns specify the interacting molecules, while the optional third and fourth columns specify colors and names/sources for the interaction, respectively. The information provided in the third and the fourth column can be used to group interactions. The color-interaction name/source combinations, when specified, are displayed in the legends panel. An example kinase-substrate interactions file (for case study II), that specifies interaction sources has been added to the GitHub repository: [https://github.com/soumitag/multiSLIDE/blob/master/demo data/kinase substrate network with sources annotated.txt](https://github.com/soumitag/multiSLIDE/blob/master/demo%20data/kinase%20substrate%20network%20with%20sources%20annotated.txt). **Fig. R1** shows a visualization generated using this file. The multiSLIDE manual (Section 7.1) has also been updated to include a description of network files as well as this visualization. In the manuscript, this functionality has been described in the *Input Data* subsection within the *Software Implementation* section.

Figure R1. Protein kinase interactions (case study II) visualized with sources tagged. This visualization uses the following network file: https://github.com/soumitag/multiSLIDE/blob/master/demo_data/kinase_substrate_network_with_sources_annotated.txt. The legends list the annotations, grouped by color and source.

For the web interface, Demo 3 (miRNA vs Protein), with everything as default:

3. What is the criterion for showing $FDR > 0.01$ ("Enter a number between 0.01-1.0")? Is it that no test is deemed statistically significant at $FDR < 0.01$, or is this imposed by the webtool?

Response: We thank the reviewer for spotting this error. In the previous version of multiSLIDE, the help text mistakenly specified the lower limit of the range as 0.01. However, the input text box was still accepting floating point values between 0 and 1 and produced results with the specified FDR thresholds. We have revised the help text in the current version to the following: "Enter a number greater than 0 and less than 1".

4. Changing FDR to 0.01, 0.05, or 0.1 leads to a bug: "Error in GlobalMapConfigServices undefined". It would be important to indicate if this is because no statistically significant test is present at these FDR, or if this is a bug in the webpage.

Response: We have attempted to reproduce this error, but we could not re-create it. We have further tested the current version of multiSLIDE to ensure that this feature produces results as expected. For all three

demo analyses, we have experimented with various FDR values, including those specified by the reviewer. In case the FDR threshold is too stringent and no data is retrieved, no heatmap is rendered and the information panel (below the omics type/title) indicates the selection criteria and that 0 genes are selected. Please let us know if this error persists.

5. The “Show neighbours > miRNA targets” does not result in any neighbours for the following miRNAs even though an edge is drawn with the protein data (“miRNA Targets of MIRNA Found 0 neighbors, of which 0 are in the datasets.”): hsa-let-7g-5p, let-7-5p/98-5p; hsa-let-7a-5p, let-7-5p/98-5p; hsa-mir-301b-3p, mir-130-3p/301-3p/454-3p.

Response: We thank the reviewer for identifying this issue. This was indeed a bug in the previous version. We have rectified this in the current version and miRNA target search now shows the neighbours when available in multiSLIDE’s internal database.

In demo 3, the interaction edges between omics types are drawn from the uploaded inter-omics file which contains a subset of connections extracted from TargetScan [1]. Independent of the user uploaded interactions, multiSLIDE’s internal database also includes connections from TargetScan. The connections identified using *Show Neighbours* are retrieved from multiSLIDE’s internal databases. Therefore, in this particular demo instance, a subset of the uploaded omics-connections overlaps with the TF Targets identified through *Show Neighbors Search*, as shown for hsa-mir-130a-3p in **Fig. R2**.

In addition, we have updated **Fig. 4** of the manuscript, by adding the pathway “metabolic pathways” to the visualization and by using a more stringent FDR threshold ($p\text{-value} \leq 0.05$, FDR 5%). Also, in the previous version, we recognized that some of the miRNA’s had been tagged to incorrect pathways, we have rectified this in the current version.

Figure R2. miRNA-protein targets of hsa-mir-130a-3p identified through neighborhood search overlap with externally uploaded TargetScanMap edges.

Reviewer #2

Comments to authors

In their manuscript the authors present an interactive web-based tool called multiSLIDE for visualization of multi-omics data and the interconnections between different omics modalities. Visualization is carried out with heatmaps where e.g. normalized expression values for genes and proteins can be shown side by side, where interconnections are highlighted with links between the heatmaps. The tool comes with many options to customize the visualizations, such as inclusion of genes/features based functional annotations, differential testing, enrichment analysis and custom lists. multiSLIDE has a knowledge base of known interactions across omics-modalities and it is possible for users to upload their own. The web interface also allows users to change the way heatmaps are displayed, e.g. how samples and features are ordered.

The manuscript is well written and comprehensive. Three case studies on published data have been carried out to illustrate how multiSLIDE can be used to explore patterns and relations in multi-omics data. The analysis methodology is sound and the results clearly demonstrate the usefulness of the tool. The work is novel, though multiSLIDE it is an extension of the authors' previously published tool SLIDE, which is applied to single-omics data (Ghosh et al. 2019).

Ghosh, S., Datta, A., Tan, K. & Choi, H. SLIDE--a web-based tool for interactive visualization of large-scale--omics data. *Bioinformatics* 35, 346–348 (2019).

A strength of multiSLIDE is that it can be used for very different types of (processed) omics data. In one of the case studies the authors provide an interesting and intuitive way to explore proteome/phosphoproteome data, e.g. by visualizing proteins and their phosphosites, or kinases and their substrates. This makes multiSLIDE powerful exploration tool for multi-omics studies.

Major revisions

The enrichment analysis option for adding genes appears non-functional. This was tested on the demo data available in <http://137.132.97.109:56695/multislide/#/>. The following error was generated: Enrichment Analysis parameters update FAILED. Could not open analysis, perhaps session has expired. Error in `do_enrichment_analysis()`.

Response: We thank the reviewer for identifying this bug. This was caused by a version mismatch between two components of multiSLIDE and the error has been fixed in the current version. We apologize for the unavailability of this feature in the previous version and encourage the reviewer to explore it in the current version.

To test the upload of custom data, the miRNA and protein data for case study II and III were separately uploaded to multiSLIDE. To visualize heatmaps, three and four terms were added with the Search function, respectively. Case study II: DNA replication, cell-cell communication and complement cascade. Case study III: phospholipid metabolism, inflammatory response pathway, glucose transport and lipid homeostasis. However, when clicking Apply changes, the following error message was issued for both studies: Changes could not be applied. Error in `AnalysisReInitializer.undefined`.

Response: We thank the reviewer for identifying this problem. We realized that searching for GO Terms without using the “go” keyword was causing this error in the previous version of multiSLIDE.

For case study II, in the previous version of multiSLIDE, we were able to visualize the data using the following three queries: “go: dna replication”, “cell-cell communication”, and “complement cascade”.

We have fixed this issue in the current version, and are able to visualize the data without using the “go” keyword as well.

The same is true for case study III, where glucose transport and lipid homeostasis are GO terms. We have tested the current version of multiSLIDE using the four pathways specified by the reviewer, and validated that search and visualization work using both keyword search and without keyword search.

In multiSLIDE, it is possible to search in a more efficient manner by providing a keyword such as “go” or “pathway”. Search queries with keywords can either take the form keyword=terms for exact search, or keyword:terms for inexact/approximate search. We have described multiSLIDE’s search syntax in Section 3.1, “Query using keyword search”, of the User Manual available here:

https://github.com/soumitag/multiSLIDE/blob/master/multiSLIDE_User_Manual.pdf

Additional details about how the enrichment analysis is performed should be provided. Which hypothesis test is used and does multiSLIDE rely on ranking genes by enrichment score (as in Gene Set Enrichment Analysis) or not? What is used as background for the test? Provided that multiSLIDE accepts pre-processed data, it is more appropriate to use all features available in the data as background, rather than the whole genome.

Response: We thank the reviewer for this useful suggestion. We have updated the following text in the *Query Mechanism* sub-section within the *Software Implementation* section of the manuscript (lines 489-501):

multiSLIDE uses hypergeometric test to evaluate statistical significance of function enrichment in the differentially expressed molecules. To perform enrichment analysis, the user specifies a set of parameters, such as the phenotypic groups to compare and identify differentially expressed genes, the significance level, to name a few. The differential expression analysis performed prior to enrichment analysis requires the same user inputs and follows the same methodology as the significance-based filtering feature described in the *Visualization Workflow* section. The background list used for the hypergeometric test includes all the molecules present in the selected omics dataset. The results of enrichment analysis are listed in the interface in an ascending order of p-values, where the user can click on enriched pathways or GO terms and add them to the visualization. Several options such as size of pathways, the number of differentially expressed genes in the pathway, are also available to filter enrichment analysis results. Users can also download a detailed report of the enrichment analysis results in a tabular format.

Minor revisions

First sentence in introduction line 31: “... technologies have made the joint use of omics...”

Response: We have shortened the sentence as follows: “Constant evolution of omics technologies such as massively parallel sequencing and mass spectrometry has made multi-omics analysis a routine practice in cell biology and clinical applications.”

Results line 94: The authors mention that, if a search generates too many hits, multiple testing correction could be applied. This correction is imperative to control false discovery rate, and the decided to perform it should not be based on number of hits obtained/wanted. Ideally, multiSLIDE should always perform correction by default.

Response: We have updated the manuscript (lines 89-92). Here we clarified that the FDR control by the BH procedure does not control the FDR throughout the whole data set, but it does so for the currently

visualized features only. As the major goal of the tool is to visualize all data and to allow the user to prioritize features by keyword searching, we believe the user should be able to see features without any type I error control for particular hypothesis tests.

Supplementary figures S2: The authors' interpretation of the heatmaps suggest that the transcriptomics profile of late time points (> 8 hours) returns to levels at time 0. While this may be true for some genes, the majority of them appear blue/red after 30 h, implying that the expression is different from 0 h.

Response: We agree with the reviewer on this point. We have revised the sentence as follows: “peaking in the intermediate phase before returning to original levels or stabilizing to a new equilibrium state different from 0h in the late phase.”

Results line 175-177: The authors mention here that a 2-hour time gap is observed between the changes in transcript and protein profiles. In Figure 2a, this pattern can be observed for some genes (notably the top ones in the heatmap). This is not as clear for those in the middle, and for several ones the transcript and protein levels appear anti-correlated.

Response: Similar to the previous comment, we agree that our description was not sufficiently detailed and accurate. We have rephrased the sentence specifically pointing out the top ER stress genes (lines 168-173).

Method line 367: “right-click” should be “left-click” to show network neighbors

Response: We have changed the text to “left-click”.

Method line 375: “left side of heatmaps” should read “right side of heatmaps”

Response: The text has been corrected to “right side of heatmaps”.

The paper could be strengthened further by including a brief conclusion.

Response: We have added another paragraph describing the novelty of work and providing future outlook on potential improvement of multi-omics data visualizations.

Web interface

Cogwheel in Synchronize Heatmap Features appears unresponsive.

Response: This has been rectified in the current version. Clicking the cogwheel now opens a panel where the user can specify which datasets/omics types should be synchronized. Un-synchronizing a dataset will re-order its features by independently clustering them. Even if a dataset/omics type is selected to be synchronized, it will only be synchronized if it has a linker column (a molecular identifier that can be mapped to Entrez or that is common between datasets).

View – Show Cluster Labels and View – Reset to Default Settings seem unresponsive.

Response: These functions have been enabled in the current version. We have integrated Show Cluster Labels feature in the global settings panel. Cluster labels are visualized only if features are ordered by hierarchical clustering and only for those datasets that are independently clustered and do not have nested linkers. In order to avoid additional tags alongside the heatmaps as well as to avoid sensory overload due to more colors, we visualize the cluster labels using two complementary color schemes for the molecular identifiers/names. A preset number of clusters are visualized, with identifiers in each subsequent cluster rendered in complementary background and foreground colors, as shown in **Fig. R3** below.

Figure R3. Five clusters visualized for the ER stress dataset.

The number of clusters to visualize can be specified in the hierarchical clustering parameters panel; that can be opened by clicking the cogwheel next to the *Order Genes By* drop down menu when *Hierarchical Clusters* is selected in the drop-down menu.

We have also enabled the *Reset to Default Settings* function, which is available under the *View* menu. This resets all configurable parameters, global and heatmap specific, to their default values, which are the same as the values these parameters have when the data is first loaded.

It is possible to add genes etc. to feature lists, but it is unclear how these are used.

Response: We thank the reviewer for highlighting this problem. We have added options to view, download and modify Feature Lists in the current version of multiSLIDE. With these features, users can curate multiple lists of interesting molecules as they explore the data and can download each list as a separate text file. Clicking the *Feature Lists* option under the *View* menu now opens a panel where all user created lists are displayed. Clicking the list name, shows the molecules within the list and clicking the download icon next to the list name, downloads a text file containing the molecules. Individual molecules as well as entire lists can be removed from the collections by clicking the respective delete buttons.

Supplementary data

README.md is confusing, it does not seem to be in agreement with the demo datasets supplied. The README does not conform to the Code and Software Submission Checklist provided with the manuscript.

Response: We agree with the reviewer that the previous README.md was confusing. We would like to highlight that the README.md file in the root directory of the GitHub repository does not describe the datasets. A separate README.md file describing the datasets is available inside the demo_data folder of the GitHub repository: https://github.com/soumitag/multiSLIDE/tree/master/demo_data. We have updated this README.md file with additional details to clearly specify how each file is used in the demo analyses.

Figure legends

Literature citations are not written in superscript.

Response: We corrected this error.

In the Fig 3 legend, PhosphoSitePlus and PhosphoNetworks are cited with references 30 and 31, but should be 31 and 32, respectively.

Response: We corrected this error.

In the Fig 4 legend, TargetScan is cited with reference 38, which appears to be a typo. The reference used in the method section is 50.

Response: We corrected this error.

References

1. Agarwal, V., Bell, G. W., Nam, J.-W. & Bartel, D. P. Predicting effective microRNA target sites in mammalian mRNAs. *Elife* **4**, (2015).

Reviewers' Comments:

Reviewer #1:

Remarks to the Author:

The authors addressed all my concerns very well. I recommend for publication.

Good luck

Ujjwal Neogi

Karolinska Institute, Sweden

Reviewer #2:

Remarks to the Author:

Comments to authors

The authors have made a considerable effort to revise and improve both the web tool and the manuscript. All my previous concerns have been addressed and the web tool features such as enrichment analysis, and searching for ontology terms and pathways, work as expected. I made some final tests of the web tool and everything worked smoothly, except for two issues outlined below.

An error was encountered when adding miRNAs/proteins in Case study III. The terms added were glycerophospholipid metabolism, glucose transport and lipid homeostasis. To clarify, I've attached a screenshot of the panel before applying changes. The error message generated was

```
Changes could not be applied. Error in AnalysisReInitializer.Exception in Selection.init():  
Exception in HttpClientManager.doGet(). Illegal character in query at index 116:  
http://127.0.0.1:5000/do_clustering?n_clusters=5&linkage_strategy=complete&group_by=0015758_  
goid,glycerophospholipid  
metabolism_pathid,0006954_goid,wp453_pathid,hsa00010_pathid,0055088_goid,r-hsa-  
70153_pathid,hsa01100_pathid,r-hsa-1430728_pathid,r-hsa-71291_pathid,r-hsa-  
556833_pathid,wp2011_pathid,0042632_goid&leaf_ordering=optimal&n_cols=31&transpose=False&di  
stance_metric=euclidean&request_id=req_1611139349565.1305374344&n_rows=193
```

Could this be due to the space inside "glycerophospholipid metabolism_pathid"?

The feature to add genes to lists was also tested with Case study I and it is possible to add individual genes to a list (new or existing). When I tried to add a gene group to a new gene list though, I initially got the following error:

```
Add feature(s) FAILED.. Exception in getGeneGroup() of FilteredSortedData.java: unknown gene  
group undefined
```

A list was created but was empty in View – Feature Lists. Similarly, adding a gene group to an exist list has no effect, even though the message says genes have been added. After retrying, it seems the error did not return, but no gene groups could be added. It also appears that deleted gene lists still show up when left-clicking on features in the heatmap and selecting either a new or existing list.

Benjamin Ulfenborg

lipid homeostasis

pathway:lipid homeostasis

abc transporters in **lipid homeostasis** [18,0]srebf and mir33 in cholesterol and **lipid homeostasis** [18,0]
goterm:lipid homeostasis

lipid homeostasis [116,0]**lipid homeostasis** [54,0] (miRNA)
phospholipid homeostasis [25,0] (miRNA)

phospholipid homeostasis [9,0]

cellular sphingolipid homeostasis [3,0]

metabolism [2035, 281]

metabolism of amino acids and derivatives [328, 0]

metabolism of lipids and lipoproteins [728, 85]

srebf and mir33 in cholesterol and lipid homeostasis [18, 3]

cholesterol homeostasis [75, 25]

inflammatory response [751, 168]

inflammatory response pathway [32, 13]

glycolysis / gluconeogenesis - homo sapiens (human) [67, 30]

glucose transport [45, 1]

metabolic pathways [281, 0]

glycerophospholipid metabolism [139, 0]

glucose transport [118, 0]

lipid homeostasis [116, 0]

APPLY CHANGES

Point-by-point response

Reviewer #2 (Remarks to the Author):

Comments to authors

The authors have made a considerable effort to revise and improve both the web tool and the manuscript. All my previous concerns have been addressed and the web tool features such as enrichment analysis, and searching for ontology terms and pathways, work as expected. I made some final tests of the web tool and everything worked smoothly, except for two issues outlined below.

An error was encountered when adding miRNAs/proteins in Case study III. The terms added were glycerophospholipid metabolism, glucose transport and lipid homeostasis. To clarify, I've attached a screenshot of the panel before applying changes. The error message generated was

```
Changes could not be applied. Error in AnalysisReInitializer.Exception in Selection.init():  
Exception in HttpClientManager.doGet(). Illegal character in query at index  
116: http://127.0.0.1:5000/do\_clustering?n\_clusters=5&linkage\_strategy=complete&group\_by=0015758\_  
goid,glycerophospholipid  
metabolism\_pathid,0006954\_goid,wp453\_pathid,hsa00010\_pathid,0055088\_goid,r-hsa-  
70153\_pathid,hsa01100\_pathid,r-hsa-1430728\_pathid,r-hsa-71291\_pathid,r-hsa-  
556833\_pathid,wp2011\_pathid,0042632\_goid&leaf\_ordering=optimal&n\_cols=31&transpose=False&dis  
tance\_metric=euclidean&request\_id=req\_1611139349565.1305374344&n\_rows=193
```

Could this be due to the space inside “glycerophospholipid metabolism_pathid”?

Response:

This error was caused by the space in the URL, as identified by Reviewer 2. Our database design requires pathway IDs in our internal databases to not contain any spaces. However, through this bug we found that a small number of pathway IDs in our default databases had spaces. We have updated the databases to rectify this and tested that these pathways can be searched and added to the visualization.

The feature to add genes to lists was also tested with Case study I and it is possible to add individual genes to a list (new or existing). When I tried to add a gene group to a new gene list though, I initially got the following error:

```
Add feature(s) FAILED.. Exception in getGeneGroup() of FilteredSortedData.java: unknown gene group  
undefined
```

A list was created but was empty in View – Feature Lists. Similarly, adding a gene group to an exist list has no effect, even though the message says genes have been added. After retrying, it seems the error did not return, but no gene groups could be added. It also appears that deleted gene lists still show up when left-clicking on features in the heatmap and selecting either a new or existing list.

Response:

We thank the reviewer for identifying this error. We have fixed this error and improved the “Add Gene Group To List” functionality. In our previous implementation, the “Add Gene Group To List” option was available in a context menu that is shown upon clicking the Feature Identifiers (Names). However, a feature can belong to multiple gene groups and this approach did not clearly mention which of these gene groups should be added to the selected list. To address this, the “Add Gene Group To List” option has

now been moved to a context menu that is available upon clicking the colored gene tags. As each gene tag is associated with a specific gene group, users will be able to accurately specify which gene group they want to add to the selected list.

Additionally, we have also improved the “View Feature List” option to more closely reflect the visualization from which the user adds the features. In the previous implementation, opening the details of a particular feature list showed each feature as an entrez ID and the list of associated aliases. However, for some datasets, such as microRNA data, the feature identifiers visualized may be neither entrez IDs nor gene names. In such cases, it can be difficult for users to correlate features in the list with features that are visualized. In our current implementation, we have changed this functionality in a way that opening the details of a particular feature list shows each feature using the matching identifiers used in the visualization. The dataset (omics-type) from which the feature was added is also noted next to the feature. The entrez ID and aliases of features, when available, can still be viewed by downloading the feature list as a delimited text file, by clicking the download list button.

Reviewers' Comments:

Reviewer #2:

Remarks to the Author:

The authors have thoroughly addressed my remaining concerns on the web tool and made additional improvements in the user interface. I carried out as some additional testing and I'm very satisfied with the tool's functionality and performance.

Benjamin Ulfenborg

Point-by-point response

Reviewer #2 (Remarks to the Author):

The authors have thoroughly addressed my remaining concerns on the web tool and made additional improvements in the user interface. I carried out as some additional testing and I'm very satisfied with the tool's functionality and performance.

Benjamin Ulfenborg

Response:

We appreciate the reviewer's careful evaluation of the implementation in both rounds of review.